# Development of Lyophilised Eudragit^®^ Retard Nanoparticles for the Sustained Release of Clozapine via Intranasal Administration

**DOI:** 10.3390/pharmaceutics15051554

**Published:** 2023-05-21

**Authors:** Rosamaria Lombardo, Marika Ruponen, Jarkko Rautio, Carla Ghelardini, Lorenzo Di Cesare Mannelli, Laura Calosi, Daniele Bani, Riikka Lampinen, Katja M. Kanninen, Anne M. Koivisto, Elina Penttilä, Heikki Löppönen, Rosario Pignatello

**Affiliations:** 1Laboratory of Drug Delivery Technology, Department of Drug and Health Sciences, University of Catania, Viale A. Doria 6, 95125 Catania, Italy; 2School of Pharmacy, University of Eastern Finland, Yliopistonranta 1C, 70210 Kuopio, Finland; marika.ruponen@uef.fi (M.R.); jarkko.rautio@uef.fi (J.R.); 3Department of Neuroscience, Psychology, Drug Research and Child Health (NEUROFARBA)—Pharmacology and Toxicology Section, University of Florence, 50139 Florence, Italy; carla.ghelardini@unifi.it (C.G.); lorenzo.mannelli@unifi.it (L.D.C.M.); 4Platform of Imaging, Department of Experimental & Clinical Medicine, University of Florence, 50139 Florence, Italydaniele.bani@unifi.it (D.B.); 5A. I. Virtanen Institute for Molecular Sciences, University of Eastern Finland, 70210 Kuopio, Finland; riikka.lampinen@uef.fi (R.L.); katja.kanninen@uef.fi (K.M.K.); 6Brain Research Unit, Department of Neurology, School of Medicine, University of Eastern Finland, 70200 Kuopio, Finland; anne.koivisto@hus.fi; 7Department of Neurology, Neuro Centre, Kuopio University Hospital, 70210 Kuopio, Finland; 8Department of Neurology and Geriatrics, Helsinki University Hospital and Neurosciences, Faculty of Medcine, University of Helsinki, 00290 Helsinki, Finland; 9Department of Otorhinolaryngology, University of Eastern Finland, Kuopio University Hospital, 70210 Kuopio, Finland; elina.penttila@pshyvinvointialue.fi (E.P.);; 10NANOMED—Research Centre on Nanomedicine and Pharmaceutical Nanotechnology, University of Catania, Viale A. Doria 6, 95125 Catania, Italy

**Keywords:** Eudragit RL100, Eudragit RS100, nanomedicine, cryoprotectants, mucoadhesion, stability

## Abstract

Clozapine (CZP) is the only effective drug in schizophrenia resistant to typical antipsychotics. However, existing dosage forms (oral or orodispersible tablets, suspensions or intramuscular injection) show challenging limitations. After oral administration, CZP has low bioavailability due to a large first-pass effect, while the i.m. route is often painful, with low patient compliance and requiring specialised personnel. Moreover, CZP has a very low aqueous solubility. This study proposes the intranasal route as an alternative route of administration for CZP, through its encapsulation in polymeric nanoparticles (NPs) based on Eudragit^®^ RS100 and RL100 copolymers. Slow-release polymeric NPs with dimensions around 400–500 nm were formulated to reside and release CZP in the nasal cavity, where it can be absorbed through the nasal mucosa and reach the systemic circulation. CZP-EUD-NPs showed a controlled release of CZP for up to 8 h. Furthermore, to reduce mucociliary clearance and increase the residence time of NPs in the nasal cavity to improve drug bioavailability, mucoadhesive NPs were formulated. This study shows that the NPs already exhibited strong electrostatic interactions with mucin at time zero due to the presence of the positive charge of the used copolymers. Furthermore, to improve the solubility, diffusion and adsorption of CZPs and the storage stability of the formulation, it was lyophilised using 5% (*w*/*v*) HP-β-CD as a cryoprotectant. It ensured the preservation of the NPs’ size, PDI and charge upon reconstitution. Moreover, physicochemical characterisation studies of solid-state NPs were performed. Finally, toxicity studies were performed in vitro on MDCKII cells and primary human olfactory mucosa cells and in vivo on the nasal mucosa of CD-1 mice. The latter showed non-toxicity of B-EUD-NPs and mild CZP-EUD-NP-induced tissue abnormalities.

## 1. Introduction

Schizophrenia is a psychiatric disorder that affects more than 20 million people worldwide [1]. The biochemical profile shows alterations in various neurotransmitter systems such as serotonergic, cholinergic, glutamatergic, GABAergic and dopaminergic. The latter seems to be the most affected, causing alterations in dopamine production that are responsible for symptoms [2]. Pharmacological treatment of schizophrenia involves the administration of typical/first-generation antipsychotics that act at the dopaminergic receptor level, i.e., by blocking dopamine D2 receptors. Extrapyramidal side effects (EPSE) reduce patient compliance and lead to discontinuation of therapy, together with the appearance of treatment resistance in 25–33% of patients [3]. In these cases, replacement therapy involves the administration of clozapine (CZP), as it is the only effective drug in drug-resistant schizophrenia and does not have EPSE [4].

CZP is a benzodiazepine and an atypical/second-generation antipsychotic that, unlike first-generation agents, has a weak affinity for the D2 receptor, where it exerts an antagonistic action reducing positive symptoms, and a greater predilection to bind the 5HT2A and D4 receptors with an antagonistic action and consequent reduction in negative symptoms [5,6]. CZP has several limitations in terms of side effects; however, the benefits are greater [7].

However, the drug dosage forms and routes of administration of CZP in clinical use show significant limitations. CZP can be administered orally (oral tablets, orodispersible tablets, suspensions), enterally (nasogastric, nasojejunal, nasoduodenal, gastrostomy and jejunostomy) and parenterally (intramuscular). Oral administration shows low bioavailability (27–47%) due to an extensive first-pass metabolism by CYP1A2 and CYP3A4 metabolic enzymes. CZP also possesses a very low solubility in water [8], and this could be a further reason for its low oral bioavailability. The time to peak plasma concentration (Tmax) is quite high (2.5 h), with an 8 h half-life following a single administration. On the other hand, the i.m. route is often painful with nodule formation, with low patient compliance and requiring specialised personnel. This route of administration is usually chosen when either the patient refuses drug treatment or when, for medical reasons, a drug cannot be taken orally [5].

This study proposes the intranasal (IN) route of administration for encapsulated CZP in polymeric nanoparticles (NPs) based on Eudragit^®^ RS100 and Eudragit^®^ RL100 copolymers (EUD). Eudragit^®^ RS100 (RS) and RL100 (RL) are copolymers of poly(ethyl methacrylate, methyl methacrylate and chlorotrimethyl ammonium methacrylate), marketed by Evonik Industries AG (Germany). They differ in the amount of quaternary ammonium groups, which is higher in RL (~10%) than in RS (~5%) [9]. The presence of ammonium salt groups allows water molecules to penetrate the polymer matrix more freely, forming a swellable matrix that can lead to a controlled/prolonged drug release that can be customised by mixing these two copolymers [10,11,12,13]. Furthermore, the salts give them a net positive charge, which is important to ensure a greater electrostatic interaction with the negatively charged mucin, which is abundant in the nasal cavity [14].

The produced NP suspensions were lyophilized, as there is some experimental evidence suggesting that a drug encapsulated in particles in the form of nasal powder has a better diffusion and absorption through the mucous membranes and shows higher bioavailability than liquid formulations. In addition, powder formulations have good storage stability and allow higher dosages of the drug to be administered [15].

The IN route, in addition to being non-invasive and easy for self-medication, with improved patient compliance, allows bypassing the first-pass effect and a more rapid onset of action. The main drug absorption route from the nose is via nose epithelia in the nasal cavity, where the drug is released from solid NPs followed by dissolution and absorption into the blood circulation through relatively leaky and highly vascularized nasal epithelia. In order to reach the brain, the drug needs to penetrate through the very tight blood–brain barrier [16]. Some studies have also suggested that small NPs, with a size under 300 nm, could be picked up by the olfactory nerve and arrive directly at the brain, bypassing the blood–brain barrier [17,18]. In this study, however, we formulated polymeric NPs with dimensions suitable for a slow release of CZP inside the nasal cavity.

Furthermore, the use of mucoadhesive polymers was aimed at increasing the residence time of NPs in the nasal cavity, as increased mucoadhesion reduces mucociliary clearance [19], resulting in improved absorption of the released drug and its bioavailability.

Thus, the paper reports experimental data on NP formulation, physicochemical characterization and freeze-drying studies. Furthermore, in vitro assays were performed in simulated nasal fluid (SNF) to assess the release kinetics of CZP from NPs, their potential interaction with mucin and NP stability. Finally, a preliminary assessment of the cytotoxicity of the systems was carried out in vitro on MDCKII cells and human primary olfactory mucosa cells, and in vivo on the nasal mucosa of CD-1 mice.

## 2. Materials and Methods

### 2.1. Materials

CZP was purchased from VWR International (Radnor, PA, USA). RS and RL are manufactured by Evonik Industries AG (Essen, Germany) and were kindly provided by Rofarma srl (Milan, Italy). Tween^®^ 80 (polyoxymethylene sorbitan monooleate) (Fluka) was used as a surfactant. Ethanol, acetone, mucin (mucin from pig stomach type II), mannitol, glucose, trehalose and sucrose are Sigma-Aldrich products purchased from Merk Life Science S.r.l. (Milan, Italy). Hydroxypropyl-β-cyclodextrin (HP-β-CD) was obtained from Roquette Freres (Lestrem, France). Tetrahydrofuran (THF), acetonitrile (ACN), dimethyl sulfoxide (DMSO) and methanol (MeOH) were purchased from VWR International Oy (Helsinki, Finland). Triethylamine and formic acid were purchased from Fluka-Garantie. SNF was composed of sodium chloride (VWR International Oy, Helsinki, Finland), potassium chloride (Fluka Chemika, Buchs, Switzerland) and calcium chloride (Riedel-de-Haën, Seelze, Germany).

### 2.2. Preparation of Unloaded RS/RL NPs (B-EUD-NPs) and RS/RL NPs Loaded with CZP (CZP-EUD-NPs) by Solvent Deposition Method

B-EUD-NPs were prepared following a described solvent deposition method [20]. An organic phase was obtained by solubilizing 100 mg of RS and RL (1:1 by weight) in 20 mL acetone. This phase was dripped in 40 mL of a water/ethanol mixture (1:1, *v*/*v*) containing 0.5% (*w*/*v*) Tween^®^ 80 (Table 1) through a burette connected to a Teflon catheter, under magnetic stirring (300 rpm) at room temperature. The formulation was then left to stir (500–600 rpm) for 30 min at room temperature for better homogenization. The organic solvents were removed under vacuum at 40 °C; the copolymers’ concentration in the final suspension was 0.5% (*w*/*v*).

CZP-EUD-NPs were prepared by the same method as above, by adding 5 mg of drug to the organic phase (final concentration: 0.025%, *w*/*v*) (Table 1).

### 2.3. Determination of Mean Particle Size (Z-Ave), Polydispersity Index (PDI) and Zeta Potential (ZP)

Photon Correlation Spectroscopy (PCS) was used to measure the mean particle size (Z-ave), polydispersity index (PDI) and Zeta potential (ZP) of NP suspensions, using a Zetasizer Nano S90 instrument (Malvern Instruments, Malvern, UK) at a detection angle of 90 °C, with a 4 mW He-Ne laser operating at 633 nm and at 25 °C. The analysis of a sample consisted of 3 sets of measurements, and the results were reported as mean size ± standard deviation (SD). Each sample was analysed using sizing cuvettes (DTS 0012). To calculate the ZP values, the software used Smoluchowski equation, whose constant has a value of 1.5, including the average values of NP mobility. ZP values are reported as the mean of three sets of up to 100 measurements.

### 2.4. Purification Method

The obtained system was purified by centrifugation with a Thermo Scientific SL 16R Centrifuge (Thermo Scientific Inc., Waltham, MA, USA) at 12,000× *g* for 1 h and at 8 °C. The obtained supernatant was collected and the pellet was resuspended in water and centrifuged again at 10,000× *g* for 30 min at 2 °C. After this washing, NPs were finally resuspended in water.

### 2.5. Lyophilization and Cryoprotection Studies

NP suspensions were mixed with 5 or 10% (*w*/*v*) of HP-β-CD and different carbohydrates to evaluate the grade of cryoprotection during the freeze-drying process. The carbohydrates tested were: mannitol, glucose, trehalose and sucrose, each one at 5, 10, 20, 30 and 40% (*w*/*v*). The mixtures were frozen at −80 °C for 24 h and then freeze-dried at a pressure of 0.300 mbar for 24 h using a Buchi Lyovapor L-200 apparatus.

The lyophilized NPs were redispersed with water in the same original volume of the aqueous nanosuspension by very mild vortex agitation. The resultant nanosuspensions were analysed by PCS to evaluate the size, PDI and ZP values.

### 2.6. FT-IR Spectroscopy Measurements

FT-IR spectroscopic analysis was performed to identify the functional groups of the substances used in the formulation and also to detect any structural changes in the polymer and/or chemical interactions that might have occurred within the nanocarriers between the drug and the polymer.

Pure CZP, pure (commercial) RS and RL resins, unloaded EUD-NPs and CZP-EUD-NPs were analysed using FT-IR spectrophotometer (PerkinElmer Spectrum RX I, Waltham, MA, USA). Each of these samples was mixed with anhydrous KBr and compressed into 1 mm disks, and the background was acquired from a pure KBr disk. The instrument was provided with an attenuated total reflectance (ATR) attachment and a diamond/zinc selenide (diamond/ZnSe) crystal window. The results were derived from 20 scans acquired in the range of 400–4000 cm^−1^ with a resolution of 2 cm^−1^ at room temperature.

### 2.7. Differential Scanning Calorimetry (DSC)

DSC analysis was performed using a DSC 2500 instrument with an RCS90 cooling unit (TA instruments, Newcastle, DE, USA) under nitrogen flow as purge gas at 50 mL/min. Briefly, 2 mg of each sample was placed in 40 µL aluminium pans, which were hermetically sealed after sample preparation. The lids of the pans were manually punctured only before each measurement. Moreover, an empty pan was used as a reference.

DSC analyses were carried out on individual raw materials (CZP, RS and RL), physical mixtures (p.m.) of CZP with RS/RL (1:1 by weight) (made by 5 mg drug and 100 mg copolymers), unloaded EUD-NPs and loaded systems (CZP-EUD-NPs). The DSC studies of the individual substances were conducted in order to examine the respective melting point (Tm) and the degree of amorphisation of the drug inside the polymer matrix.

The samples were equilibrated at 15 °C, then heated from 15 °C up to 300 °C at a rate of 5 °C/min. Thermograms were processed and analysed using the TA instruments Trios software v5.1.1.46572.

#### Stability Study of CZP-EUD-NPs Powder by DSC Analysis

The DSC analysis was also used to analyse the stability of CZP-EUD-NPs powder stored for up to 8 months in a vacuum-free desiccator containing silica gel at a room temperature (25 ± 3 °C) and at 50% relative humidity.

### 2.8. High Performance Liquid Chromatography (HPLC) Analysis

#### HPLC Method for the Quantification of CZP

HPLC analysis was performed using an Agilent 1100 binary pump (Agilent Technologies Inc., Wilmington, DE, USA), a 1100 micro vacuum degasser, a HP 1050 Autosampler, a HP 1050 variable wavelength detector (operated at 235 nm). The chromatographic separations were achieved on a Supelco SupelcosilTM LC-SI analytical column (4.6 mm × 250 mm, 5 μm) (Supelco Inc., Bellefonte, PA, USA) by using isocratic elution of 25 mM of phosphate buffer with 0.3% of triethylamine at pH 4.7 and ACN with 0.1% of formic acid (30:70, *v*/*v*). Effluent was monitored at a wavelength of 254.4 nm, with a flow rate of 1.5 mL/min; the injection volume was 5 µL; retention time of 3.7 min. The column was maintained at 40 °C throughout the analysis.

Standard calibration curves were prepared at different dilutions of CZP: in Milli-Q^®^ water/ACN (1:1, *v*/*v*) with a linear regression coefficient determined in the range 0.5–200 μg/mL, which was 0.9999; in DMSO with a linear regression coefficient determined in the range 0.1–200 μg/mL, which was 1.

### 2.9. Drug Loading (% DL) and Encapsulation Efficiency (% EE)

To evaluate % DL and % EE of NPs, CZP was extracted from NPs: CZP-EUD-NPs were dissolved in DMSO (1 mg/mL). The solution was centrifugated at 6000× *g* for 1 h in a centrifugal filter falcon (MW cutoff: 30 KDa). Considering that RS and RL have an average molecular weight of 150 KDa [9], the filtrate was analysed using HPLC-UV following the method condition previously described. The experiments were repeated in triplicate and the results are reported as mean ± standard deviation.

% DL was calculated according to the following formula:$$\%\;DL=\frac{Current\;drug\;quantity\;\left(mg\right)}{Quantity\;of\;system\;\left(mg\right)}\times 100$$

% EE was calculated according to the following formula:$$\%\;EE=\frac{Current\;drug\;quantity\;\left(mg\right)}{Theoretical\;drug\;quantity\;\left(mg\right)}\times 100$$

### 2.10. In Vitro Studies

#### 2.10.1. Stability Studies

##### Stability Study of CZP-EUD-NPs Stored at 25 °C and 4 °C

Unloaded and loaded NPs were stored at 25 °C and 4 °C. Their stability was monitored for 90 days, examining size and PDI by using a Zetasizer Nano S90 instrument (Malvern Instruments, Malvern, UK) under the same conditions previously described.

##### Stability Studies in Simulated Nasal Fluid (SNF)

To evaluate the influence of the ions, which are in SNF, and the mucin on the stability of lyophilized unloaded and loaded NPs, they were resuspended in water or in SNF (composition: 0.877 g NaCl, 0.058 g CaCl_2_ and 0.298 g KCl in 100 mL of double distilled water; pH 5.5) with or without 0.1% (*w*/*v*) of mucin. Samples were incubated at 35 ± 0.5 °C for 1 and 24 h, after which Z-ave and PDI were checked to evaluate any aggregation phenomenon.

#### 2.10.2. Release Study of CZP from NPs by Using Rapid Equilibrium (RED) Inserts

The in vitro release profile of CZP from NPs was conducted using the RED inserts, which have 8000 Da molecular weight cutoff (Thermo Fisher Scientific, Waltham, MA, USA). NPs (1 mg) were suspended in SNF (1.25 mL) at pH 5.5 and incubated at 35 ± 0.5 °C under slight stirring (80 rpm) for 8 h.

During the whole assay time, aliquots of 0.5 mL were withdrawn periodically and analysed by HPLC-UV with the previously described method. The same amount of subtracted fluid was replaced with fresh artificial medium, which was added back to the initial incubated system to restore the original volume and sink conditions. The experiment and HPLC-UV analyses were repeated in duplicate.

#### 2.10.3. Release Kinetic Analysis

The in vitro release data of CZP-loaded NPs were analysed according to various kinetic models:
Zero-order model: R = K_o_tFirst-order model: R = 1 − e^−kt^Higuchi model: R = K_H_ t^1/2^Hixson–Crowell model: W_o_^1/3^ − W_t_^1/3^ = K_HC_tKorsmeyer–Peppas model: R = k_KP_ t^n^


The amount of the drug released at time t; k_o_, k, k_H_, K_HC_ and k_Kp_ (k are rate constants for zero-order, first-order, Higuchi, Hixson–Crowell and Korsmeyer–Peppas kinetic models, respectively) is expressed by R. W_o_ is the initial amount of drug and W_t_ is the amount of drug at time t; n is the release exponent in the Korsmeyer–Peppas model [21]. The model with the highest correlation coefficient (R^2^) was selected to describe and evaluate the mechanism of CZP release. Calculations were made on the linear part of the release curves (from 1 h forward).

#### 2.10.4. Mucoadhesion Studies

The interactions between mucin and NPs were studied to evaluate the potential mucoadhesive proprieties of NPs by two in vitro methods: turbidimetry and zeta potential.

##### Mucoadhesion Evaluation of Unloaded and Loaded NPs by Turbidimetric Assay

The evaluation of the potential mucoadhesive properties of unloaded and loaded NPs was performed by a turbidimetric assay with a modification of the method of He et al. [22], using a Genesis 10 w Scanning ThermoElectron UV spectrophotometer (Waltham, MA, USA), a quartz cuvette with an optical path length of 10 mm and operating at λ = 650 nm [14,18,23,24].

Briefly, mucin (0.1%, *w*/*v*) was suspended in SNF (pH 5.5) and shaken overnight to allow complete dispersion. CZP-EUD-NPs and unloaded NPs were suspended in equal volumes of the mucin dispersion and shaken for 15 min at 25 °C. Samples were incubated at 35 ± 0.5 °C for 1 or 24 h and then submitted to UV analysis. The absorbances of blank or drug-loaded NPs, incubated in SNF under the same above-described conditions, and mucin in SNF were measured as controls. SNF was used in the reference cuvette.

Turbidity values were calculated according to the following formula:$$\Delta Abs=A-A_{\mathrm{theor}}$$

The effective absorbance (*A*) of the NPs–mucin mixtures was compared to the theoretical absorbance (*A*_theor_) calculated by adding the individual absorbance values of 0.1% mucin and NP suspensions in SNF. The difference (Δ*Abs*) was taken as a measure of the interaction between mucin and NPs [22]. If Δ*Abs* ≤ 0, it means that there is no interaction; if Δ*Abs* > 0, a strong interaction between mucin and the NPs is envisaged [25]. The experiments were repeated in duplicate.

##### Mucoadhesion Evaluation of CZP-EUD-NPs by Zeta Potential (ZP)

The evaluation of the potential electrostatic interaction between mucin and NPs was performed by PCS to measure ZP.

CZP-EUD-NPs and unloaded NPs were suspended in equal volumes of mucin (0.1% *w*/*v*) and shaken for 15 min at 25 °C. Samples were then incubated at 35 ± 0.5 °C for 0, 1 and 24 h. The control samples were prepared by resuspending NPs in water and incubating under the same above conditions.

The ZP of each sample was analysed by a Zetasizer Nano S90 instrument (Malvern Instruments, Malvern, UK) as previously described (Section 2.4). The experiments were repeated in duplicate.

### 2.11. Toxicological Evaluation of Nanoparticles

#### 2.11.1. In Vitro Studies

##### Cell Culture

MDCKII was purchased from Netherlands Cancer Institute (Amsterdam, Netherlands). Cells were cultured in Dulbecco’s modified Eagle medium (DMEM) supplemented with 10% (*v*/*v*) fetal bovine serum (FBS), 1% penicillin–streptomycin and 1% L-glutamine. The cells were grown at 37 °C in 5% of CO and used in the experiment in passages 26–30.

A piece of olfactory mucosa (OM) was collected as a biopsy from the nasal septum, near the roof of the nasal cavity under the ethical approval from the Research Ethics Committee of the Northern Savo Hospital District (permit number 536/2017) from one cognitively healthy individual. Primary OM cell cultures were set up and cultured according to the previously published protocol [26,27,28]. The culture medium contained DMEM/F12 supplemented with 10% (*v*/*v*) FBS, 1% penicillin–streptomycin (all reagents obtained from Gibco, Waltham, MA, USA). The cells were grown at 37 °C in 5% of CO_2_ and used in the experiment as primary passage 7.

##### Cell Viability Assay

Cell viability was evaluated by using 3-[4,5 dimethylthiazol-2-yl]-2,5-diphenyltetrazolium bromide (MTT) assay (Sigma-Aldrich, St. Louis, MO, USA). MDCKII cells were seeded into 96-well plates at a density of 100,000 cells/mL in 200 μL of supplemented growth medium. OM cells were seeded into 48-well plates at a density of 20,000 cells/mL in 250 μL of supplemented growth medium. After 24 h for MDCKII cells and around 72 h for OM cells, they were exposed to B-EUD-NPs and CZP-EUD-NPs at concentrations 2 mg/mL, 1 mg/mL and 0.5 mg/mL in six and eight replicates, respectively, for 24 h. Cells not exposed to NPs and cultured in a growth medium were used as control. Then, the cells were incubated for 2 h with 10% of MTT (0.5 mg/mL) in a serum-free medium for MDCKII cells and a medium with 10% of serum for OM cells. Therefore, 100 µL of SDS-DMF lysis buffer for MDCKII cells and 100 µL of dimethyl sulfoxide (DMSO) for OM cells was added, to obtain cell lysis and solubilization of blue formazan crystals resulting from MTT reduction by viable cells’ activity, and incubated overnight and for 20 min, respectively. Cell viability was measured by the reduction in MTT solution. The optical density of the formed blue formazan was measured at 570 nm for MDCKII cells and 595 nm for OM cells using a Victor^2^ multilabel plate reader (PerkinElmer, Wallac, St. Paul, MN, USA). The cell viability % was calculated as below:$$\%\;of\;cell\;viability=\frac{\left(Abs\;exposed\;cells-Abs\;blank\right)}{\left(Abs\;non-exposed\;cells-Abs\;blank\right)}\times 100\;$$

#### 2.11.2. In Vivo Studies

##### Animals

CD-1 mice (Envigo, Varese, Italy) weighing 20–25 g were used to assess the toxicity of NPs in vivo. The animals were housed at the Centro Stabulazione Animali da Laboratorio (University of Florence) and were used at least one week after their arrival.

There were ten mice per cage (size 26 × 41 cm), which were fed a standard laboratory diet and tap water ad libitum and maintained at 23 ± 1 °C with a 12 h light/dark cycle (light at 7 am). The manipulations to which they were subjected were in accordance with Directive 2010/63/EU of the European Parliament and the Council of the European Union (22 September 2010) on the protection of animals used for scientific purposes. The ethical policy of the University of Florence complies with the Guide for the Care and Use of Laboratory Animals of the US National Institutes of Health (NIH Publication No. 85–23, revised 1996; University of Florence assurance number: A5278-01). Formal approval to conduct the described experiments was obtained from the Italian Ministry of Health (No. 498/2017) and the Animal Subjects Review Board of the University of Florence. Animal experiments were reported according to ARRIVE guidelines [29]. All efforts were made to minimise animal suffering and to reduce the number of animals used. The animals were randomly divided into different groups and treated with the test products. Of each product, 25 μL per nostril per day was administered for 7 days.

##### Tissue Preparation

After the treatments with CZP-EUD-NPs, mice were sacrificed by decapitation. The head was quickly dissected and the anterior part of the snout including the nasal cavities was isolated. The specimens were fixed by immersion in Immunofix (Bio-Optica, Milan, Italy) for 24 h, followed by decalcification in Biodec R demineralizing solution (Bio-Optica), freshly replaced every 2 days, until adequate tissue softening was achieved (approximately 15 days). Then, the specimens were rinsed, dehydrated in graded ethanol and embedded in paraffin. Finally, 6 μm thick sections were cut, dewaxed and stained with haematoxylin and eosin (H&E). The effects of the different treatments on the nasal mucosa were observed in order to assess: (i) the integrity of the surface ciliated epithelium, (ii) the signs of mucus overproduction, (iii) the occurrence of vasodilatation and inflammatory infiltrate.

### 2.12. Statistical Analysis

The data were expressed as mean ± standard deviation. The results were analysed using one-way ANOVA and the differences between groups were considered significant for a *p*-value < 0.05, very significant for *p*-value < 0.01 and extremely significant for *p*-value < 0.001. StatPlus^®^ software (version V8) was used for statistical analysis.

## 3. Results

### 3.1. Technological Characterization of Unloaded and Drug-Loaded EUD-NPs

B-EUD-NPs and CZP-EUD-NPs batches were prepared by a solvent deposition method. PCS analysis showed that B-EUD-NPs had the following characteristics: an average size of about 400 nm, a narrow distribution width (PdI ≤ 0.1) and a positive surface charge. The positive charge is given by the presence of ammonium groups in RS and RL copolymers. On the basis of the results obtained with the blanks, which show repeatability of the data (Table 2), the CZP-loaded polymer particles were formulated successfully.

CZP-EUD-NPs showed a size greater than unloaded NPs, which is in the range between 400 and 500 nm. This size (>300 nm) prevents NPs from being picked up by the olfactory nerve, thus allowing them to reside in the nasal cavity where the drug can be released and absorbed through the nasal mucosa and reach the systemic circulation [12]. Furthermore, loaded NPs also showed a high uniformity of distribution width (PdI ≤ 0.1) and a positive surface charge higher than unloaded NPs (Table 3).

In terms of drug encapsulation, CZP-EUD-NPs showed a % DL of 0.5 ± 0.01 and a % EE of 10.46 ± 0.13.

### 3.2. Lyophilization Process and Cryoprotection Study of NPs

The suspension of NPs obtained was lyophilised as experimental evidence shows that the drug encapsulated in particles in the form of nasal powder has better diffusion and absorption across mucous membranes and shows greater bioavailability than liquid formulations. In addition, powder formulations have better storage stability and allow higher dosages of the drug to be administered [15]. Although the freeze-drying process has the advantages described above, it may affect both the particle size, which should be unchanged, and the reconstitution of the final product, which should be easy to achieve [30].

CZP-EUD-NP samples were easily reconstituted in water after freeze-drying (AF) by 4–5 min of gentle agitation at 200 rpm and analysed by PCS. They showed a larger size than before freeze-drying (BF) (ratio of final to initial size: Sf/Si = 1.74; *p*-value ≤ 0.001) due to some aggregation phenomena. To stabilise the size of NPs during the lyophilization process, cryoprotectants such as HP-β-CD, mannitol, glucose, trehalose and sucrose were added to the NP suspension. There are several scientific studies showing that they possess a good degree of cryoprotection against polymeric NPs [31,32,33,34,35]. They create a protective shell, a glassy/vitreous coating, around the NPs to prevent aggregation caused by the mechanical stress of the ice crystals [36]. Furthermore, the degree of cryoprotection depends not only on the type of cryoprotectant but also on its concentration. For the coating of the glassy matrix, which forms around the NPs, to be complete, the required cryoprotectant concentration should be identified. The PDI indicates whether the amount of cryoprotectant used is sufficient, and in this case, it will be as close as possible to that of the non-lyophilized formulation. In the case of the PDI value being high, the amount of cryoprotectant is in excess because it does not deposit on the surface of the NPs and forms different populations. For this reason, the above-mentioned carbohydrates were tested at different concentrations, from 5 to 40% (*w*/*v*); HP-β-CD was instead tested at 5 and 10% (*w*/*v*).

PCS analysis of the reconstituted specimens revealed that of all the potential cryoprotectants used, sucrose at 10% (*w*/*v*) and HP-β-CD at 5 and 10% (*w*/*v*) showed the best cryoprotection with Sf/Si values of 1.06, 0.98 and 0.89, respectively (Table 4).

There was a significant difference in the size of the NPs lyophilized with cryoprotectant compared to those dried without cryoprotectant (*p*-value ≤ 0.001). Notably, NPs cryoprotected with HP-β-CD showed a highly significant difference in PDI compared to those dried without any cryoprotectant (*p*-value ≤ 0.001). In contrast, the sample lyophilized with 10% (*w*/*v*) sucrose showed no significant difference in PDI.

Furthermore, comparing the mean size and PDI of the NPs-BF with the cryoprotected and freeze-dried samples, it can be observed (Figure 1) that in the presence of 5% (*w*/*v*) HP-β-CD, a significant difference in either size or PDI was not registered (*p*-value ≤ 0.05), whereas when using 10% (*w*/*v*) HP-β-CD or sucrose, a very significant difference (*p*-value ≤ 0.001) was noted. Thus, among the tested cryoprotectants, HP-β-CD at 5% (*w*/*v*) showed the most efficient cryoprotective action for CZP-EUD-NP: the size (523.1 nm) and PDI (0.110) values of the reconstituted suspensions were closely correlated with those (531.9 nm and 0.103, respectively) of the same formulation before lyophilization. The ability of HP-β-CD to keep the NPs well separated may be due to their immobilization inside the glassy matrix formed by the hydrogen bond between the cyclodextrin and polar groups on the NP surface. In particular, previous studies have shown that the cyclic structure of these oligoglucoside compounds appears to perform better adsorption on the NP surface during vapour sublimation [37].

### 3.3. Infrared and Thermal Analysis of NPs

CZP showed a characteristic band at 3303.8 cm^−1^, related to the free N-H stretching of the diazepine ring [38]; the bands at 2990.1 and 2929.2 cm^−1^ were due to aliphatic C-H stretching; the bands at 1618 and 1557 cm^−1^ were due to C=N stretching vibrations, while those at 1500 and 1471.7 cm^−1^ were attributable to aromatic C=C stretching vibrations. CZP also showed a characteristic band due to C-Cl that extended to about 796.8 cm^−1^ [39].

The IR spectrum of pure RS showed a peak at 1735.5 cm^−1^, characteristic of C=O ester elongation vibrations. The peak at 3435.5 cm^−1^ was assigned to the quaternary amine stretching vibration, while the peak at 2937.0 cm^−1^ corresponded to the aliphatic stretching vibration [40]. Similarly, RL showed a peak at 1738.3 cm^−1^ for the C=O stretching vibration of the ester group; the peak at 3430.0 cm^−1^ was assigned to the quaternary amine stretching vibration, while the peak at 2944.3 cm^−1^ corresponded to the aliphatic stretching vibration [41].

Unloaded B-EUD-NPs showed an IR spectrum identical to that of the pure copolymers; CZP-EUD-NPs displayed a similar pattern, although some peaks related to the encapsulated drug were visible, for instance around 1620–1630 cm^−1^, suggesting that CZP was present in the nanoparticles but dispersed within the polymer network and only to a low extent located on the NP surface (Figure 2), a behaviour also reported in other studies [31,42].

DSC analyses were performed on individual raw materials (CZP, RS and RL), the physical mixture (p.m.) of RS/RL and CZP, unloaded nanoparticles (B-EUD-NPs) and drug-loaded ones (CZP-EUD-NPs). DSC studies of the individual substances were conducted to evaluate the melting points (Tm) and the degree of crystallinity or amorphisation of CZP dispersed in the polymer matrix.

The calorimetric curve of pure CZP showed a strong endothermic peak, respectively, at 184.42 °C, which corresponds to Tm and indicates their crystalline nature [43]. The same peak was found in the physical mixture at a lower intensity, proportional to the amount of CZP present, while it was not visible in CZP-EUD-NPs, indicating that the drug could be dispersed in an amorphous state in the polymeric matrix (Figure 3) [31,33].

DSC analyses were carried out on a system stored for 8 months in a vacuum-free desiccator containing silica gel at a room temperature of 25 ± 3 °C and 50% relative humidity. The calorimetric curve shows that CZP-EUD-NPs are very stable as no signs of degradation are present (data available from the Authors).

### 3.4. In Vitro Studies

#### 3.4.1. Stability Studies

Unloaded and loaded NP suspensions were stored at 25 °C and 4 °C for 90 days. PCS analysis showed the size and PDI of the EUD-B and EUD-C systems remained constant over time (1, 2, 15, 30, 60, 90 days) at both temperatures, the size being in the range of 400 nm and the PDI with a value mostly ≤0.1. The results show that the systems were very stable and no aggregation phenomena occurred. Furthermore, the suspensions showed a clear appearance that did not change throughout the study (Figure 4 and Figure 5).

To study the behaviour of lyophilized NPs after IN administration and the influence of ions present in SNF and mucin, they were resuspended in water and SNF with 0.1% (*w*/*v*) of mucin and incubated at 35 ± 0.5 °C for 1 and 24 h.

PCS analysis showed that the size and PDI of the B-EUD-NPs and CZP-EUD-NPs suspended in water and incubated at 35 ± 0.5 °C were stable after 1 h. Although after 24 h of incubation the PDI values had increased, the size did not show any increase that could indicate the formation of aggregates (Figure 6).

Formulations suspended in SNF with 0.1% (*w*/*v*) mucin and incubated at 35 ± 0.5 °C already showed a significant increase in size after 1 h, which further increased after 24 h. This is confirmed by the increase in PDI, which indicates the presence of different populations (Figure 7). This increase in size and PDI values can be explained by the presence of mucin, which, having a net negative charge, interacted electrostatically with the polymeric NPs characterised by a net positive charge [14].

#### 3.4.2. Release Study of CZP from NPs by Using Rapid Equilibrium (RED) Inserts and Kinetic Analysis

An in vitro release profile of CZP from EUD-NPs was carried out using RED inserts by reproducing the physiological environment of the nasal cavity, i.e., in SNF incubated at 35 ± 0.5 °C under stirring at 80 rpm. The sink condition was maintained throughout the release studies.

Figure 8 shows the release profile of CZP from EUD-NPs, which exhibited complete release within 8 h (99.85%). After the first hour during which the drug closer to the particle surface was rapidly released, RS created a thicker layer that slowed the entry of the surrounding medium into the NPs; therefore, CZP release was slower and was sustained along the remaining time. On the other hand, this action was balanced by the presence of RL (1:1 by weight), which possesses more quaternary ammonium groups than RS and allows a greater erosion and permeability of the polymer matrix to the surrounding medium [44], thus leading to a complete drug release in 8 h.

The release rate constant was estimated from the slope of the different curves, and regression values (R^2^) were obtained. Table 5 shows that the in vitro CZP release from EUD-NPs was best described by the first-order equation (R^2^ = 0.9992), suggesting that drug release was dependent on its concentration [45].

#### 3.4.3. Mucoadhesivity Evaluation

The epithelium of the nasal mucosa is lined with glandular cells that secrete mucus as a protective mechanism for the trapping and elimination of xenobiotic substances. Therefore, to prevent NPs from being quickly eliminated due to mucociliary clearance and thus increase their residence time in the nasal mucosa, it is possible to increase the viscosity of the formulation, and use mucopenetrators or bioadvised materials that form NPs, e.g., the hydrophilic polymers. After coming into contact with mucus, the polymer chains undergo hydration, i.e., swelling, and become entangled with the mucin fibers. This entanglement is caused by the formation of bonds such as electrostatic attractive forces, hydrophobic interactions, hydrogen bonds and van der Waals bonds. The formation of these bonds promotes not only a longer residence time of the NPs in the nasal mucosa but also an increase in the contact time with the mucosa, which promotes greater absorption and thus higher bioavailability of the drug [46].

RS and RL copolymers have shown potential mucoadhesive properties [14]. Due to the presence of the positive net charge from the ammonium groups, they can develop electrostatic interactions with mucin, which possesses a negative net charge due to the presence of sialic acid and sulphate residues.

In this study, loaded and unloaded NPs were suspended in equal volumes of 0.1% (*w*/*v*) of mucin dispersion in SNF (pH 5.5), and after incubation at 35 ± 0.5 °C for 0, 1 and 24 h, they were submitted to a turbidimetric analysis. The difference in absorbance between blank and loaded samples (ΔAbs) was taken as a measure of the interaction between mucin and NPs.

The turbidimetry assay confirms the potential mucoadhesive properties of RS and RL copolymers, the main constituents of B-EUD-NPs and CZP-EUD-NPs. In particular, neither systems showed an interaction with mucin at zero time because the ΔAbs values were ≤0. However, B-EUD-NPs and CZP-EUD-NPs showed interactions with mucin after 1 and 24 h of incubation, with ΔAbs values > 0. According to statistical analysis, B-EUD-NPs exhibited a stronger interaction than CZP-EUD-NPs after 1 h (*p* ≤ 0.001), whereas after 24 h, there was no statistical significance between the two systems (Figure 9).

An interpretation of these findings can be related to the presence of CZP in drug-loaded samples. The drug in fact seems to negatively affect the interaction with mucin, both at the initial time, when it could even cause a temporary disaggregation of mucin chains, as reported in the literature under other conditions [47], and after 1 h, when the ΔAbs value was almost null. Such an effect can be related to the positive charge borne by CZP molecules, whose pka is around 7.5–7.8, at the pH value of SNF (5.5). After 24 h, probably because of the concomitant release of the encapsulated drug, the loaded NPs showed the same behaviour as that of blank carriers in this experiment.

ZP assessment was carried out to compare the data from the turbidimetric study. In addition, in this assay, B-EUD-NPs and CZP-EUD-NPs were suspended in equal volumes of 0.1% (*w*/*v*) mucin dispersion in SNF (pH 5.5) and incubated at 35 ± 0.5 °C for 0, 1 and 24 h. Figure 10 shows that both systems already exhibited a considerable reduction in ZP at the zero time point (*p* ≤ 0.001). The positive charge of the polymer chains could be neutralised by the negative charges of mucin, with a consequent change in ZP towards neutrality. These data are in accordance with those reported in Figure 7, which show a size increase in the NPs due to the interaction with the protein. However, this experiment, unlike the turbidimetric assay, shows that the interaction was immediate because ZP values had already suffered a considerable reduction at time zero. After 1 and 24 h of incubation, the ZP value of both systems continued to decrease. In particular, statistical analysis showed that B-EUD-NPs interacted more strongly than CZP-EUD-NPs after 0, 1 and 24 h (*p* ≤ 0.001), confirming to some extent the adverse effect of CZP upon the interaction with mucin.

### 3.5. Cell Viability of MDCKII and OM Cells Treated with Nanoparticles

NPs proposed for IN administration must be able to deliver and release CZP into the nasal cavity without compromising host cell viability. To assess whether B-EUD-NPs and CZP-EUD-NPs can induce cytotoxicity, the MTT assay was performed on MDCKII cells and primary OM cells. The MTT study was conducted on two different cell types to assess whether cell viability was strictly dependent on the type of cell and concentration tested.

NPs were tested at different concentrations (0.5–2 mg/mL) to assess the effect on cell viability and the potential application of these systems as nanocarriers via IN. Figure 11 shows the cell viability of MDCKII cells and OM cells versus NP concentration (mg/mL).

Regarding MDCKII cells incubated for 24 h with B-EUD-NPs and CZP-EUD-NPs, they showed high viability (>87%) in the concentration range 0.5–1 mg/mL. At the highest NP concentration (2 mg/mL), cell viability was slightly reduced (70% for CZP-EUD-NPs and 67% for B-EUD-NPs).

Statistical analysis of the data showed that the decrease in viability compared to the control at the NP concentration of 2 mg/mL was highly significant (*p* < 0.001) for B-EUD-NPs and for CZP-EUD-NPs. At concentrations of 1 mg/mL, B-EUD-NPs showed an extremely significant (*p* < 0.001) reduction in viability compared to the control, while CZP-EUD-NPs exhibited no significant difference. At lower concentrations, i.e., 0.5 mg/mL, they showed no significance in the reduction in viability compared to the control.

OM cells incubated for 24 h with B-EUD-NPs showed a viability of 75% in the NP concentration range 0.5–2 mg/mL. In contrast, OM cells incubated for 24 h with the same concentrations of CZP-EUD-NPs showed a viability ≥50%.

Statistical analysis of the data showed that the decrease in viability compared to the control for both systems at a concentration of 2 mg/mL was highly significant (*p* < 0.001). At a concentration of 1 mg, B-EUD-NPs showed a very significant difference (<0.01) and CZP-EUD-NPs exhibited an extremely significant difference (*p* < 0.001). Moreover, both systems showed a very significant difference (*p* < 0.01) at a concentration of 0.5 mg.

According to ISO 10993-5, percentages of cell viability greater than 70% are considered to show the absence of cytotoxicity [48]. Thus, it can be stated that the loaded and unloaded NPs showed no signs of cellular cytotoxicity in MDCKII cells at all the tested concentrations (0.5, 1 and 2 mg/mL), as they gave cell viability percentages of 70% or more, with the exception of CZP-EUD-NPs, which at the concentration of 2 mg/mL, showed a slightly lower viability (66%).

In contrast, drug-loaded NPs showed signs of cell cytotoxicity in OM cells at concentrations of 1 and 2 mg/mL, with a viability around 50%. At a lower NP concentration (0.5 mg/mL), viability was around 70%. Such findings would of course deserve a deeper investigation to determine the role of drug loading on cell cultures.

### 3.6. Histopathological Analysis

To evaluate the potential toxicity of unloaded and loaded NPs in nasal mucosa after IN administration, 50 μL suspension of NPs was IN administered two times per day for one week. Figure 12 shows the results of the histological analysis. Unloaded B-EUD-NPs did not cause detectable morphological alterations: in particular, the septal and lateral mucosa showed normally appearing ciliated columnar epithelium, muciparous glands and stroma, with no signs of mucus hyperproduction, blood vessel dilation and inflammatory infiltrate. On the other hand, CZP-EUD-NPs did induce slight, albeit appreciable, tissue abnormalities, consisting of hypertrophy of the muciparous glands and increased amounts of mucus and exfoliated cells in the nasal cavities; the stromal components appeared normal.

## 4. Conclusions

In this work, the preparation of lyophilised NPs made by Eudragit^®^ Retard copolymers was developed for IN administration to allow a prolonged residence and release of CZP in the nasal cavity, with the aim of improving drug bioavailability at a systemic level.

The NPs showed dimensions in the 400–500 nm range, high uniformity of distribution (PdI ≤ 0.1) and a positive surface charge due to ammonium groups. In order to protect and preserve these characteristics from the freeze-drying process, cryoprotection studies were conducted, showing that 5% (*w*/*v*) HP-β-CD not only ensures the complete resuspension of the lyophilizate upon reconstitution but also preserves the size and PDI of the NPs.

Solid-state characterisation studies of the loaded NPs confirmed that the drug was encapsulated and homogeneously dispersed in the polymer matrix and was only to a low extent located on the NP surface. Furthermore, DSC analysis demonstrated the stability of the dry systems for up to 8 months. Stability studies were also performed on the aqueous suspensions at 4 °C and 25 °C for 90 days, showing that the systems were very stable and no particle aggregation occurred. The stability of NPs lyophilised in water and SNF added with 0.1% mucin was evaluated to assess the influence of the ions and glycoprotein present. The results of NPs incubated in distilled water showed that the size was stable up to 24 h, which could indicate that the ions did not induce any particle aggregation. In SNF containing mucin, the NPs already showed a significant increase in size and PDI after 1 h that further increased after 24 h. This could suggest an electrostatic interaction between the NPs and mucin, explainable with their positive surface charge. Such mucoadhesive properties are important in favouring a longer residence time on the nasal mucosa, thus improving the possibility of drug absorption into the systemic blood flow.

Results of the in vitro release studies in SNF showed that CZP-EUD-NPs are able to provide controlled CZP release for up to 8 h. The release profile had the best fit with the first-order equation (R^2^ = 0.9992); however, the Higuchi equation also fitted well with the experimental data (R^2^ = 0.998).

After IN administration, EUD-NPs, in addition to being able to release CZP inside the nasal cavity, should not impair host cell viability, as assessed on primary MDCKII and OM cell lines. NPs did not show any toxic effect in the concentration range 0.5–2 mg/mL on MDCKII cells, with the exception of CZP-EUD-NPs at 2 mg/mL, which showed a cell viability slightly below 70%. Furthermore, on OM cells, unloaded NPs at a concentration of 0.5–2 mg/mL did not show signs of cytotoxicity, as cell viability was always around 75% or above; conversely, drug-loaded NPs showed a reduction in cell viability of 50% in the concentration range 1–2 mg/mL.

Finally, in vivo studies in mice confirmed the non-toxicity of B-EUD-NPs, as they caused no detectable morphological changes. Instead, CZP-EUD-NPs induced slight, but appreciable, tissue abnormalities that deserve a deeper investigation.

## Figures and Tables

**Figure 1 pharmaceutics-15-01554-f001:**
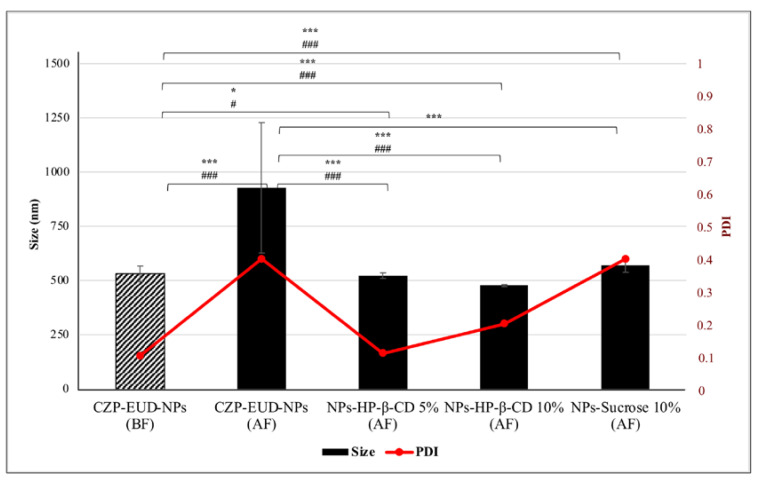
Mean size and polydispersity index (PDI) of CZP-EUD-NPs before (BF) and after freeze-drying (AF), without or with the addition of HP-β-CD at 5 and 10% (*w*/*v*) or sucrose at 10% (*w*/*v*) as cryoprotectants. Significance value of size is denoted by the symbol * and of the PDI by the symbol #. They were set as a *p*-value ≤ 0.05 (*,^#^) and ≤ 0.001 (***,^###^).

**Figure 2 pharmaceutics-15-01554-f002:**
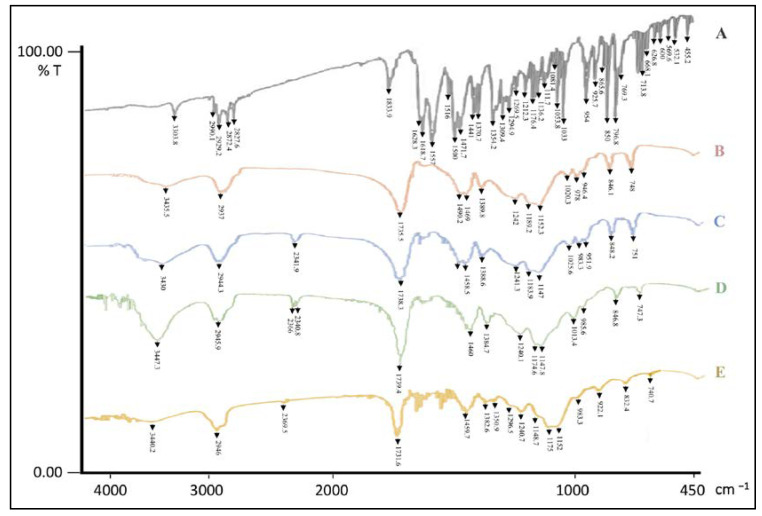
FT-IR spectra of (A) CZP, (B) commercial RS, (C) commercial RL, (D) B-EUD-NPs and (E) CZP-EUD-NPs.

**Figure 3 pharmaceutics-15-01554-f003:**
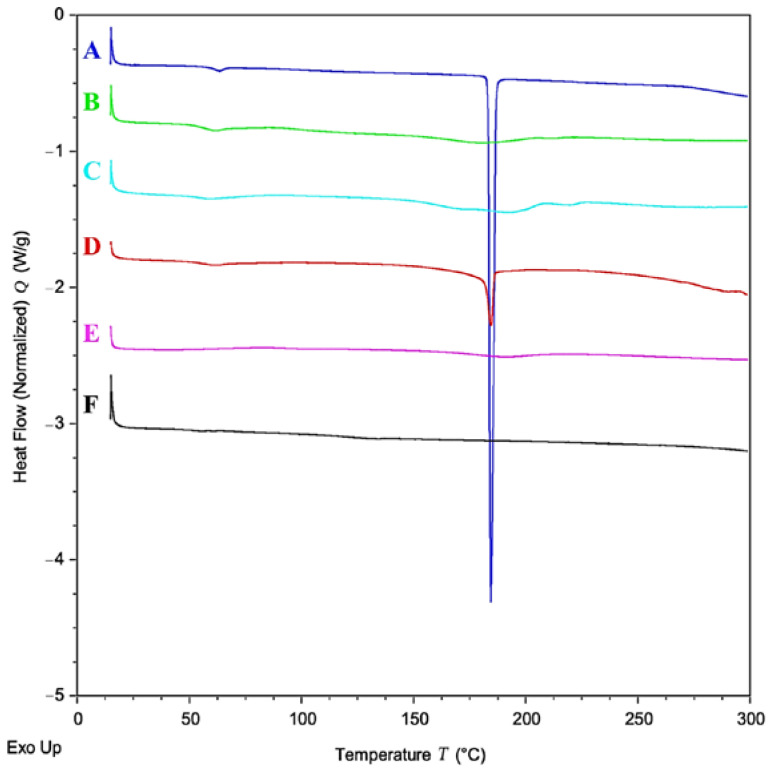
DSC thermogram of (A) neat CZP; (B) RS; (C) RL; (D) RS/RL (1:1)-CZP (10:1, *w*/*w*) physical mixture; (E) B-EUD-NPs; (F) CZP-EUD-NPs.

**Figure 4 pharmaceutics-15-01554-f004:**
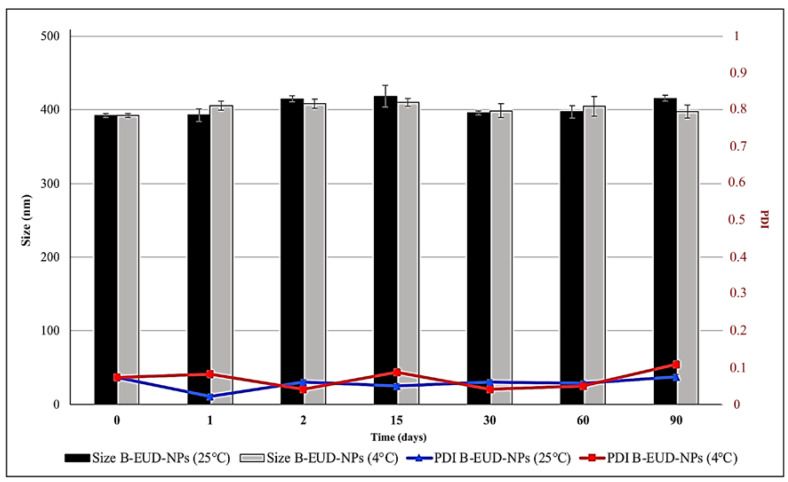
Stability studies (changes in mean size and PDI) of B-EUD-NP suspensions stored at 4 and 25 °C for 90 days.

**Figure 5 pharmaceutics-15-01554-f005:**
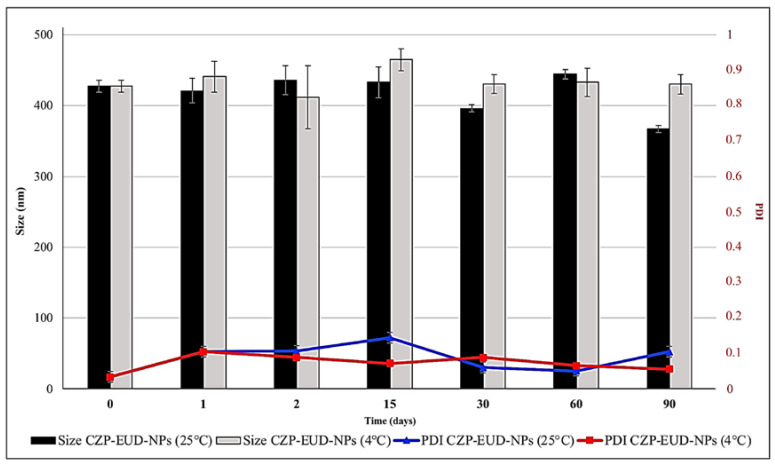
Stability studies (changes of mean size and PDI) of CZP-EUD-NP suspensions stored at 4 and 25 °C for 90 days.

**Figure 6 pharmaceutics-15-01554-f006:**
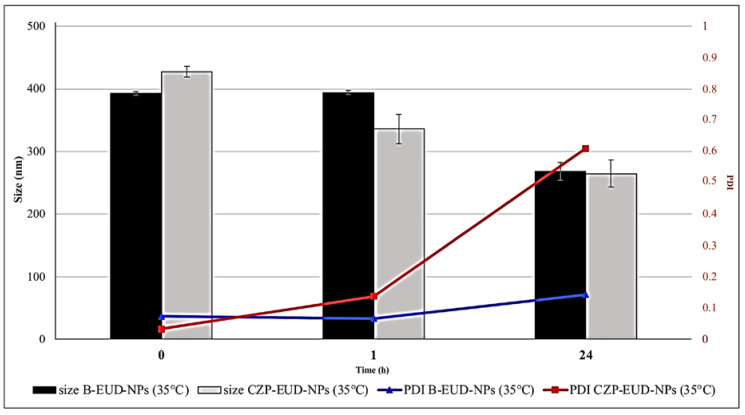
Stability studies (changes in mean size and PDI) of B-EUD-NPs and CZP-EUD-NPs incubated in water at 35 ± 0.5 °C for 0, 1 and 24 h.

**Figure 7 pharmaceutics-15-01554-f007:**
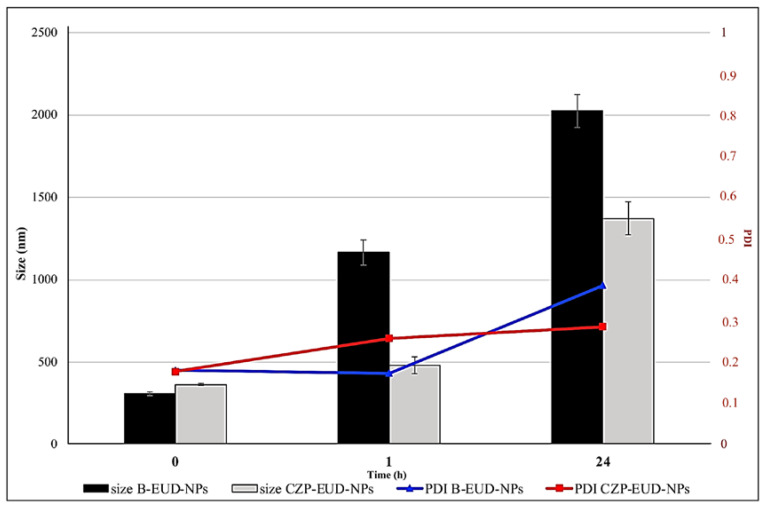
Stability studies (changes in mean size and PDI) of CZP-EUD-NPs incubated in SNF with 0.1% (*w*/*v*) mucin at 35 ± 0.5 °C for 0, 1 and 24 h.

**Figure 8 pharmaceutics-15-01554-f008:**
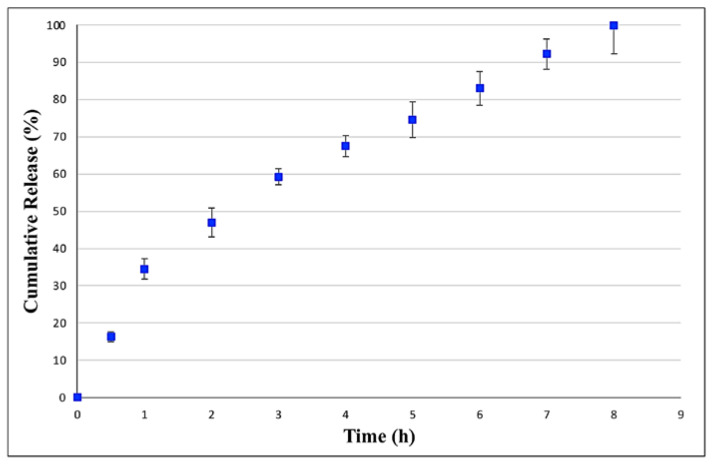
In vitro release profile of CZP from EUD-NPs in SNF (pH 5.5) at 35 ± 0.5 °C under stirring at 80 rpm.

**Figure 9 pharmaceutics-15-01554-f009:**
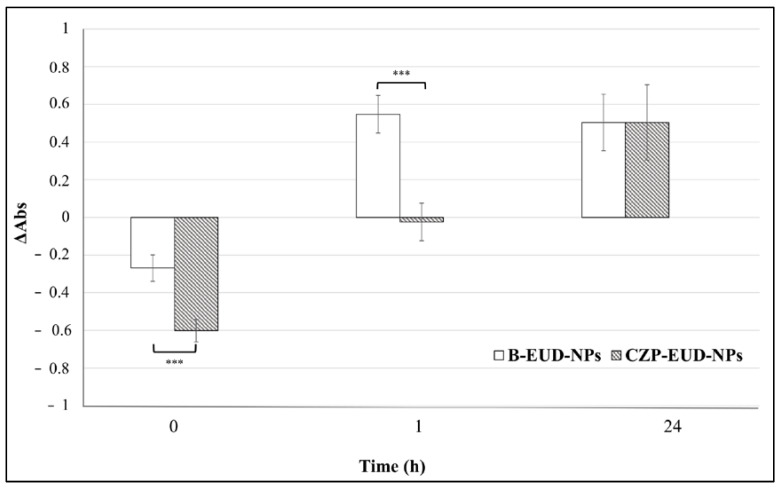
Change in ΔAbs (650 nm) of B-EUD-NPs and CZP-EUD-NPs mixed with mucin (0.1%, *w*/*v*) and incubated at 35 ± 0.5 °C in SNF at pH 5.5 for 0, 1 and 24 h. Significance was set at *** *p* ≤ 0.001.

**Figure 10 pharmaceutics-15-01554-f010:**
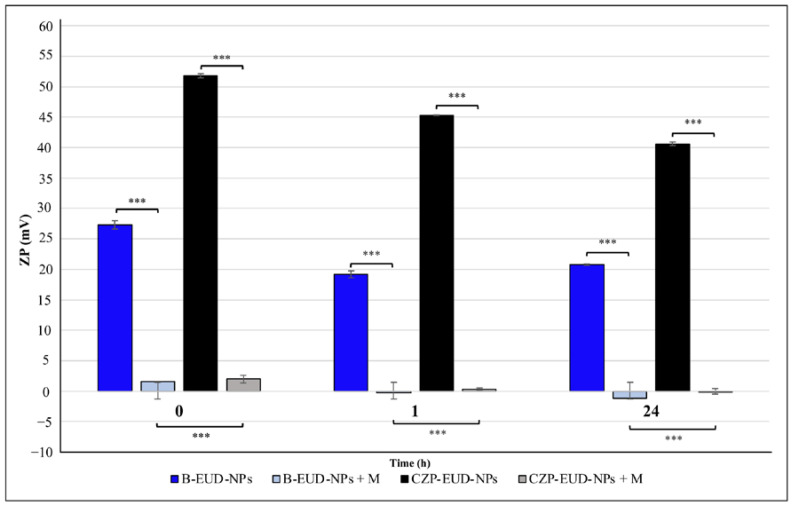
ZP values of B-EUD-NPs and CZP-EUD-NPs suspended in water and incubated at 35 ± 0.5 °C for 0, 1 and 24 h. B-EUD-NPs + M (M: mucin) and CZP-EUD-NPs + M were NP samples suspended in SNF (pH 5.5) with mucin (0.1%, *w*/*v*) and incubated at 35 °C for 0, 1 and 24 h. Significance was set at *** *p* ≤ 0.001.

**Figure 11 pharmaceutics-15-01554-f011:**
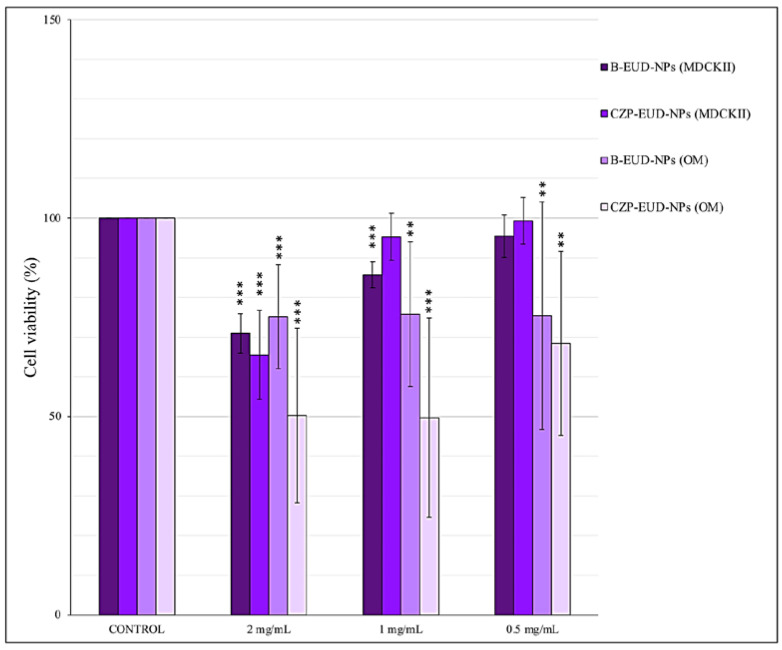
Cell viability after exposure for 24 h of Madin–Darby kidney cells (MDCKII) and human primary olfactory mucosa cells (OM) derived from two different individuals to B-EUD-NPs and CZP-EUD-NPs at 2 mg/mL, 1 mg/mL and 0.5 mg/mL in eight and six technical repeats for each, respectively. Significance was set at ** *p* ≤ 0.01; *** *p* ≤ 0.001.

**Figure 12 pharmaceutics-15-01554-f012:**
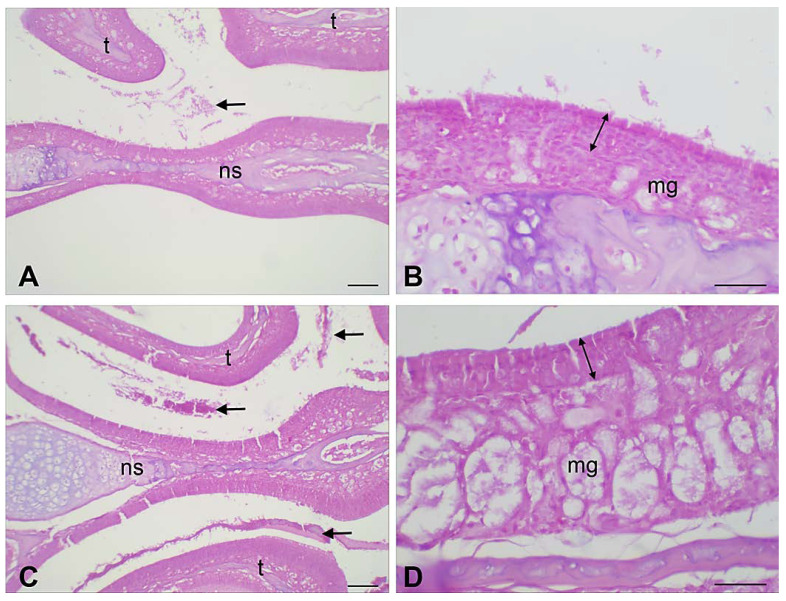
Histological features of the nasal mucosa of CD-1 mice administered with 50 μL suspension of B-EUD-NPs (**A**,**B**) and CZP-EUD-NPs twice per day for one week (**C**,**D**) (ns, nasal septum; t, turbinates; mg, muciparous glands). Arrows point at clusters of mucus and exfoliated material; double arrows mark the width of the columnar surface epithelium, including cilia. In the mucosa of CZP-EUD-NP-treated mice, muciparous gland hypertrophy and mucus hyperproduction are the most prominent changes. Haematoxylin and eosin; bars = 100 µm.

**Table 1 pharmaceutics-15-01554-t001:** Composition of NP batches.

Batch	CZP (mg)	Copolymer(s) (mg)	RL:RS Weight Ratio	Water/ EtOH Ratio	Water + EtOH Volume (mL)	Acetone Volume (mL)	Tween^®^ 80 (% *w*/*v*)
B-EUD-NPs	-	100	1:1	1:1	40	20	0.5
CZP-EUD-NPs	5	100	1:1	1:1	40	20	0.5

**Table 2 pharmaceutics-15-01554-t002:** Mean size (Z-Ave), polydispersity index (PdI) and zeta potential (ZP) of B-EUD-NPs.

Batch	Z-Ave (nm) + SD	PDI + SD	ZP ± SD (mV)
B-EUD-NPs (1)	394 ± 3	0.067 ± 0.062	27.3 ± 0.7
B-EUD-NPs (2)	392 ± 3	0.074 ± 0.050	23.6 ± 2.5
B-EUD-NPs (3)	382 ± 5	0.100 ± 0.072	26.3 ± 1.6

**Table 3 pharmaceutics-15-01554-t003:** Mean size (Z-Ave), polydispersity index (PdI) and zeta potential (ZP) of CZP-EUD-NPs.

Batch	Z-Ave (nm) + SD	PDI + SD	ZP ± SD (mV)
CZP-EUD-NPs (1)	461 ± 1	0.101 ± 0.050	51.8 ± 0.4
CZP-EUD-NPs (2)	427 ± 9	0.034 ± 0.011	50.5 ± 0.9
CZP-EUD-NPs (3)	531 ± 35	0.095 ± 0.079	53.7 ± 0.6

**Table 4 pharmaceutics-15-01554-t004:** PCS characterization (mean size and ratio of final (after lyophilization) to initial size, S_f_/S_i_) of NPs reconstituted in water after lyophilization (AF) with and without cryoprotectants.

Batch	Z-Ave (nm) + SD	S_f_/S_i_
CZP-EUD-NPs (AF)	926 ± 37	1.74
NPs- HP-β-CD 5% (AF)	523 ± 13	0.98
NPs- HP-β-CD 10% (AF)	477 ± 5	0.89
NPs- Sucrose 10% (AF)	569 ± 29	1.06

**Table 5 pharmaceutics-15-01554-t005:** Regression coefficient values (R^2^) for different release kinetic models obtained from release kinetic profile CZP-EUD-NPs. The calculations were made on the linear part of the release curves (from 1 h forward).

Zero-Order	First-Order	Higuchi	Hixson–Crowell	Korsmeyer–Peppas
0.9837	0.9992	0.9980	0.9979	0.9845

## Data Availability

The experimental data supporting the reported results are available upon request from the corresponding authors.

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
