# Peer review of "Development of Lyophilised Eudragit® Retard Nanoparticles for the Sustained Release of Clozapine via Intranasal Administration"

_pharmaceutics, 2023, doi:10.3390/pharmaceutics15051554_

Round 1

Reviewer 1 Report

The manuscript by Lombardo et al. presents an extensive and comprehensive study on the preparation, characterization and biomedical properties of clozapine loaded polymeric nanoparticles.

The study is well performed and thorougly conducted. Only minor points can be improved for easier understanding:

- what are the chemical compositions of and the differences between the two polymers? One sentence could be added in the introduction

- what was the dilution for pcs measurements? Was KCl soultion used for measuring the zeta potential?

- the ratio beween particles and lyophilization agent is unclear

- what is the role of Tween80? Is it necessary for stabilization, despite the positive zeta potential? Or is it altering surface properties for cell interaction?

- l420: the authors claim experimental evidence but do not city any literature. Please add.

- the decimal places given for the particles size are meaningless. Integers are physicocheimally mor reasonable

- resolution of IR-spectra can be improved

- l554: although it is reasonalbe to assume that interaction with mucin leads to aggregation and distinct populations, no evidence is presented. Thus this statement should be avoided or confirmed by experimental evidence

- are the fits applied for the datapoints after burst release? If yes, please add to the text

Author Response

The manuscript by Lombardo et al. presents an extensive and comprehensive study on the preparation, characterization and biomedical properties of clozapine loaded polymeric nanoparticles.

The study is well performed and thorougly conducted. Only minor points can be improved for easier understanding:

  1. what are the chemical compositions of and the differences between the two polymers? One sentence could be added in the introduction

A/ we added the following sentence:

Eudragit® RS100 (RS) and RL100 (RL) are copolymers of poly(ethyl methacrylate, methyl methacrylate and chlorotrimethyl ammonium methacrylate). They differ in the amount of quaternary ammonium groups, i.e. RL contains more of them (~10%) than RS (~5%) [9]. The ammonium groups are present as salts and allow water molecules to penetrate the polymer matrix more freely. Thus, drug release profiles can be customized by mixing these two polymers [9].

  1. what was the dilution for pcs measurements? Was KCl soultion used for measuring the zeta potential?

A/  The analysis was done without dilution because the suspension was not very torbid. KCl solution was not used.

  1. the ratio beween particles and lyophilization agent is unclear

A/  The ratio of suspension NPs to cryoprotectant was expressed as percent weight/volume.

  1. what is the role of Tween80? Is it necessary for stabilization, despite the positive zeta potential? Or is it altering surface properties for cell interaction?

A/ Using Tween derives from the previous method used for the production of Eudragit Retard nanoparticles (Quasi emulsion solvent diffusion, QESD; see the various papers from Pignatello et al.). Using the present method, we observed that the presence of a surfactant helps in the correct and uniform aggregation of the copolymers into discrete nanoparticles, during the evaporation of the org solvents.

  1. l420: the authors claim experimental evidence but do not city any literature. Please add.

A/  Reference 19 is already present in the text.

  1. the decimal places given for the particles size are meaningless. Integers are physicochemically more reasonable

A/ thank you for the remark, me made the suggested changes in the relevant tables.

  1. resolution of IR-spectra can be improved

A/ Unfortunately, it cannot be improved. We produced them in a separate lab and the files received had the quality of the present picture.

  1. l554: although it is reasonalbe to assume that interaction with mucin leads to aggregation and distinct populations, no evidence is presented. Thus this statement should be avoided or confirmed by experimental evidence

A/ The following phrase was added:

Formulations suspended in SNF with 0.1% (w/v) mucin and incubated at 35 ± 0.5 °C showed a significant increase in size already after 1 hour, which further increased after 24 hours. This is confirmed by the increase in PDI, which indicates the presence of different populations (Figure 7).

  1. are the fits applied for the datapoints after burst release? If yes, please add to the text

A/ The calculations were made on the linear part of the release curves (from 1 hour forward). We specified this in the paper.

Reviewer 2 Report

In the manuscript pharmaceutics-2319539 the lyophilized Eudragit® Retard RS100/RL 100-based nanoparticles were developed for sustained release of clozapine. The manuscript in its present form cannot be published due to poor design, lack of thorough analysis and discussion of the results. In addition, there are the following major and minor comments on the manuscript.

Major:

1. It seems that the authors themselves developed a technique for obtaining nanoparticles, and it is not clear from the manuscript on the basis of what they took certain amounts of substances and solvents, and then stirred for 30 minutes at room temperature. I ask the authors to explain the scheme for the synthesis of nanoparticles or refer to published articles.

2. What are the resulting nanoparticles? What is their shape? I ask the authors to provide TEM or SEM images for the obtained nanoparticles.

3. What type of chemical bond is responsible for incorporating clozapine into the nanoparticles? I ask the authors to prove spectra confirming inclusion of the drug. I understand that in the IR spectrum and DSC thermogram of the loaded system, no characteristic peaks of the functional groups of the drug were found. Could such a lack of a signal indicate the absence of drug inclusion in nanoparticles, but the location of the drug on the surface of nanoparticles? Is it possible to accurately determine the location of a drug with nanoparticles using NMR spectroscopy?

4. The manuscript should be supplemented with the chemical structures of RS/RL 100 polymers.

5. The manuscript states that the redispersion of samples (5 and 10% HP-β-CD and 10% sucrose) was evaluated by macroscopic observation. What does this mean and is it possible to present the results of this observation?

6. Due to what HP-β-CD at a concentration of 5% (w/v) showed the most effective cryoprotective effect?

7. In some figures with two Y axes, it is difficult to correlate the data to these axes. Can the authors somehow facilitate this correlation with certain colors?

8. In Figure 9, can authors give the value of the wavelength at which absorption was taken into account?

9. I ask the authors to provide a detailed description of the preparation of SNF.

10. Conclusions should be shortened and outline the main results and/or further recommendations. The order of conclusions should correspond to the order in which they are mentioned in the discussion of the results.

11. The entire manuscript should be structured, avoiding paragraphs consisting of two or even one sentence.

Minor:

It is necessary to give an expansion of the abbreviation HP-β-CD.

Editing needed “There is also some studies...”.

Need a comma “In order to reach brain the…”.

Author Response

In the manuscript pharmaceutics-2319539 the lyophilized Eudragit® Retard RS100/RL 100-based nanoparticles were developed for sustained release of clozapine. The manuscript in its present form cannot be published due to poor design, lack of thorough analysis and discussion of the results. In addition, there are the following major and minor comments on the manuscript.

Major:

  1. It seems that the authors themselves developed a technique for obtaining nanoparticles, and it is not clear from the manuscript on the basis of what they took certain amounts of substances and solvents, and then stirred for 30 minutes at room temperature. I ask the authors to explain the scheme for the synthesis of nanoparticles or refer to published articles.
  •  B-EUD-NPs were prepared following an already described solvent deposition method [20]. In those previous studies by our lab many preliminary studies and attempts were made to choose the better conditions.
  1. What are the resulting nanoparticles? What is their shape? I ask the authors to provide TEM or SEM images for the obtained nanoparticles.

A/  We do not have a direct access to these facilities; therefore, we can start with a SEM analysis in a separate lab, but this would take 2-3 weeks to have the results. We can suggest to go ahead with the revision and, in the case of acceptance, we will insert this morphological part in the final version of the paper.

  1. What type of chemical bond is responsible for incorporating clozapine into the nanoparticles? I ask the authors to prove spectra confirming inclusion of the drug. I understand that in the IR spectrum and DSC thermogram of the loaded system, no characteristic peaks of the functional groups of the drug were found. Could such a lack of a signal indicate the absence of drug inclusion in nanoparticles, but the location of the drug on the surface of nanoparticles? Is it possible to accurately determine the location of a drug with nanoparticles using NMR spectroscopy?

A/  Absence of the peculiar peaks of the drug in these situations is usually taken as a proof that it is uniformly dispersed (sometimes in an amorphus form) inside the polymer matrix, and not onto the particle surface. In our own experience, this is very common with Eudragit Retard micro/nanoparticles. The drug is of course present in the systems, as proven by the UV analysis after in vitro release assays.

Literature very rarely reports NMR studies to assess the drug/polymer interaction in nanoparticles or similar systems. However, we cannot perform NMR studies at this stage are requested, even because the solid state instrument would be necessary.

  1. The manuscript should be supplemented with the chemical structures of RS/RL 100 polymers.

A/ we added the following sentence:

Eudragit® RS100 (RS) and RL100 (RL) are copolymers of poly(ethyl methacrylate, methyl methacrylate and chlorotrimethyl ammonium methacrylate). They differ in the amount of quaternary ammonium groups, i.e. RL contains more of them (~10%) than RS (~5%) [9]. The ammonium groups are present as salts and allow water molecules to penetrate the polymer matrix more freely. Thus, drug release profiles can be customized by mixing these two polymers [9].

  1. The manuscript states that the redispersion of samples (5 and 10% HP-β-CD and 10% sucrose) was evaluated by macroscopic observation. What does this mean and is it possible to present the results of this observation?

A/ The entire section of ms was revised and made more understandable. The caption of Table 4 was also rewritten more clearly. Of course, freeze-dried NPs were reconstituted and analyzed by PCS vs the original samples (before freeze-drying) to discuss the effect of this technique and of the cryoprotectants on their properties.

  1. Due to what HP-β-CD at a concentration of 5% (w/v) showed the most effective cryoprotective effect?

A/  We discussed in the text this aspect:

‘The ability of HP-β-CD to keep the NPs well separated is due to their immobilization within the glassy matrix formed by the hydrogen bond between the sugar and polar groups on the NPs’ surface. In particular, previous studies have shown that the cyclic structure of these oligoglucosidic compounds appears to perform better adsorption on the NPs’ surface during water sublimation [34]’.

  1. In some figures with two Y axes, it is difficult to correlate the data to these axes. Can the authors somehow facilitate this correlation with certain colors?

A/ thank you, we changed the relevant figures.

  1. In Figure 9, can authors give the value of the wavelength at which absorption was taken into account?

A/ the information was added.

  1. I ask the authors to provide a detailed description of the preparation of SNF.

A/ SNF composition was added (0.877 g NaCl, 0.058 g CaCl2 and 0.298 g KCl in 100 mL of double distilled water; final pH: 5.5).

  1. Conclusions should be shortened and outline the main results and/or further recommendations. The order of conclusions should correspond to the order in which they are mentioned in the discussion of the results.

A/ Thank you, we made some changes in the section.

  1. The entire manuscript should be structured, avoiding paragraphs consisting of two or even one sentence.

A/ A revision of the text was made accordingly.

Minor:

It is necessary to give an expansion of the abbreviation HP-β-CD.

A/ The chemical name of this cyclodextrin was reported.

Editing needed “There is also some studies...”.

A/ change made in the ms.

Need a comma “In order to reach brain the…”.

A/ change made in the ms.

Round 2

Reviewer 2 Report

I thank the authors for correcting the manuscript, but I still have the following questions and suggestions for it:

1. I am not satisfied with the answer that the absence of peculiar peaks of the drug is usually taken as evidence that it is uniformly dispersed (sometimes in amorphous form) within the polymer matrix, and not on the surface of the particles. Since the authors submit a scientific manuscript in a first-quartile journal, this statement must be convincingly explained from a scientific point of view and supported by a literary reference from a first-quartile journal.

2. Please increase the frequency values in Fig. 2.

3. What is the reason for choosing the wavelength of 650 nm? If the authors have UV spectra, it would be nice to show them as a separate figure, because there are very few spectral results in the manuscript.

Author Response

  1. I am not satisfied with the answer that the absence of peculiar peaks of the drug is usually taken as evidence that it is uniformly dispersed (sometimes in amorphous form) within the polymer matrix, and not on the surface of the particles. Since the authors submit a scientific manuscript in a first-quartile journal, this statement must be convincingly explained from a scientific point of view and supported by a literary reference from a first-quartile journal.

A/  We rewrote the entire discussion, with a deeper analysis of the experimental IR spectra. A further reference was added, too.

  1. Please increase the frequency values in Fig. 2.

The IR graphs were extended to 450 nm.

As we already wrote, IR spectra were acquired in a separate lab, and their re-recording and/or management of the actual figures would require the production of new nanoparticle batches.

If this point of the revision will be considered crucial for the whole study, we go ahead with this operation.

  1. What is the reason for choosing the wavelength of 650 nm? If the authors have UV spectra, it would be nice to show them as a separate figure, because there are very few spectral results in the manuscript.

A/ We reported some refs. (from high IF journals), in which authors choose a wl of 650 nm to analyze the bond of NPs with mucin. Therefore, we relied on such previous experiences.

We do not think that would be necessary to add in the paper the UV spectra of mucin in saline, since literature is already rich of papers that use the same wavelength (650 nm) in this kind of experiments.

Round 3

Reviewer 2 Report

Thanks for the revisions to the manuscript. The question regarding the wavelength used for turbidimetric analysis arose because there are articles published in the journal Pharmaceutics 10.3390/pharmaceutics14010170, 10.3390/pharmaceutics15030921 in which mucoadhesion was evaluated at 500 nm. And I became interested in viewing the UV spectra of Figure 9, because I do not understand the negative change in absorbance at a wavelength of 650 nm by 0.6 units for 0 h. In addition, for sure, encapsulated drugs have their own absorption bands, which may change when included in nanoparticles. Presenting the UV spectra of the individual drug in the absence and presence of nanoparticles and still in the presence of mucin would help convince me of the confirmation of drug encapsulation, a question I raised during the first round of peer review. In this regard, I ask the authors

1. explain the negative change in absorption at a wavelength of 650 nm by 0.6 units for 0 h in Fig.9.

2. indicate the thickness of the cuvette in which the test solutions for turbidimetry were placed.

3. clarify which solution was used as a control or reference measurement for turbidimetry.

Author Response

Dear colleague, we revised the entire section of turbidimetry and mucoadhesion experiments, both in the experimental section and in R&D one (evidenced in red in the ms).

Please let us know if the discussion added could answer to your comments

regards

Rosario Pignatello

Round 4

Reviewer 2 Report

regards